# Mixing Performance of the Modified Tesla Micromixer with Tip Clearance

**DOI:** 10.3390/mi13091375

**Published:** 2022-08-23

**Authors:** Makhsuda Juraeva, Dong-Jin Kang

**Affiliations:** School of Mechanical Engineering, Yeungnam University, Gyoungsan 38541, Korea

**Keywords:** degree of mixing (DOM), modified Tesla micromixer, tip clearance, symmetric counter-rotating vortices, drag and connection of interface

## Abstract

A passive micromixer based on the modified Tesla mixing unit was designed by embedding tip clearance above the wedge-shape divider, and its mixing performance was simulated over a wider range of the Reynolds numbers from 0.1 to 80. The mixing performance was evaluated in terms of the degree of mixing (DOM) at the outlet and the required pressure load between inlet and outlet. The height of tip clearance was varied from 40 μm to 80 μm, corresponding to 25% to 33% of the micromixer depth. The numerical results show that the mixing enhancement by the tip clearance is noticeable over a wide range of the Reynolds numbers Re < 50. The height of tip clearance is optimized in terms of the DOM, and the optimum value is roughly h = 60 μm. It corresponds to 33% of the present micromixer depth. The mixing enhancement in the molecular diffusion regime of mixing, Re ≤ 1, is obtained by drag and connection of the interface in the two sub-streams of each Tesla mixing unit. It appears as a wider interface in the tip clearance zone. In the intermediate range of the Reynolds number, 1 < Re ≤ 50, the mixing enhancement is attributed to the interaction of the flow through the tip clearance and the secondary flow in the vortex zone of each Tesla mixing unit. When the Reynolds number is larger than about 50, vortices are formed at various locations and drive the mixing in the modified Tesla micromixer. For the Reynolds number of Re = 80, a pair of vortices is formed around the inlet and outlet of each Tesla mixing unit, and it plays a role as a governing mechanism in the convection-dominant regime of mixing. This vortex pattern is little affected as long as the tip clearance remains smaller than about h = 70 μm. The DOM at the outlet is little enhanced by the presence of tip clearance for the Reynolds numbers Re ≥ 50. The tip clearance contributes to reducing the required pressure load for the same value of the DOM.

## 1. Introduction

Micromixers are widely used in many microfluidic systems for biochemistry analysis, chemical synthesis, biomedical diagnostics, and drug delivery [1,2,3]. As the microfluidic systems aims to achieve several characteristics such as reduced consumption of reagent, fast processing, low cost, and portability [4], they require rapid and complete mixing. Micromixing is therefore one of the fundamental technologies utilized in microfluidic applications.

The mixing in most microfluidics systems is governed by molecular diffusion, slow fluid velocity and microscale geometry. The associated flow corresponds to very low Reynolds number regime, and mixing is inevitably slow and inefficient. Therefore, it is critical to develop a more efficient micromixer for the progress of the microfluidic industry. Mixing enhancement is still a crucial design goal, even though various technologies have been proposed to enhance the efficiency of microfluidic mixing [2].

A variety of micromixers have been proposed to enhance the mixing in microfluidic systems, and they are usually categorized as either active or passive. An active micromixer utilizes an external energy source to improve mixing efficiency. The external energy source is mostly used to generate flow disturbance, and contributes to the enhancement of mixing. Typical energy sources are acoustic [5], magnetic [6], electric [7], thermal [8], and pressure [9]. As each active micromixer employs an external energy source, the resulting structure of an active micromixer is more complicated and expensive, compared with passive micromixers. This characteristic limits the usage of active technologies in microfluidic systems. On the contrary, passive micromixers rely on the modification of geometric structures to generate a chaotic flow field and have no moving parts. Therefore, they are much simpler to integrate into a microfluidic system. Various geometric modifications have been shown to generate a chaotic flow field. Some of them include a staggered herringbone [10], channel wall twisting [11], repeated surface groove and baffles [12,13], block in the junction [14], split-and-recombine (SAR) [15,16], Tesla structure [17], stacking of mixing units in the cross-flow direction [18], optimization of lateral structure [19] and submergence of planar structures [20].

There are several approaches to enhance the degree of mixing (DOM): complex three-dimensional structures, modification of planar geometry, and manipulation of flow conditions. Using a pulsatile inlet flow is an example to control flow condition. For example, McDonough et al. [21] showed that the micromixing time decreased with an increased velocity ratio of oscillatory velocity to net velocity; baffle designs were used. However, this kind of approach requires an extra device to generate the pulsatile flow. In this paper, a simpler geometric approach is studied to enhance the mixing performance, based on a geometric modification.

Generally, a complex three-dimensional (3D) micromixer may result in a better mixing performance than that of a two-dimensional (2D) micromixer of similar size [22]. However, the entire fabrication process of a 3D micromixer is much more complicated, and costs more compared with a planar design. In addition, some 2D planar micromixers were shown to generate effective 3D flow characteristics such as multidirectional vortex and Dean vortex. For example, Hong et al. [23] proposed a modified Tesla micromixer, which is based on the Coanda effect. The Coanda effect allows the fluid to follow the angles surface, and the Tesla structure is placed serially in the opposite direction to enhance the transverse dispersion of fluid. Hossain et al. [17] optimized this modified Tesla micromixer and showed that it generates a couple of symmetric counter-rotating vortices in the cross section for the Reynolds numbers Re ≥ 2. Raza et al. [22] recommended this modified Tesla structure in the intermediate (1 < Re ≤ 40) and high Reynolds number ranges (Re > 40) and recommended 3D micromixers in the low Reynolds range (Re ≤ 1). On the other hand, Makhsuda et al. [20] showed that a submergence of planar structure enhances the mixing performance in the Reynolds number range of Re ≥ 5; a vortex burst of the two Dean vortices promotes mixing performance. Chung et al. [24] showed that short planar baffles with a gap promotes vortex creation due to the sudden expansion around the baffles and enhance the mixing performance in the diffusion-dominant flow rate (Re < 1) and the convection-dominant flow rate (Re > 40). Bazaz et al. [25] studied a hybrid micromixer combining six planar mixing units such as modified Tesla, ellipse-like, nozzle, pillar, teardrop and obstruction, and optimized the combination: one nozzle, one pillar, three obstacles in a curved channel, and two modified Tesla units. They obtained a mixing performance improvement for a wide range of Reynolds number (Re ≤ 1 and 22 ≤ Re ≤ 45).

In this paper, tip clearance was embedded into the modified 2D Tesla micromixer to enhance the mixing performance by combining the Coanda effect and flow disturbance due to tip clearance of planar structures. The present micromixer consists of several modified Tesla units, and tip clearance is present above the wedge-shape divider geometry of each modified Tesla unit. As the structure of the present micromixer is slightly modified from that of a planar micromixer, microfabrication techniques such as Xurography [26] can be easily applied. The Xurography technique uses thin, pressure-sensitive double-sided adhesive flexible films so that the tip clearance zone is simply tailored using a cutter plotter; it cuts off a film along the perimeter of the modified Tesla micromixer. The tailored film and the planar structure can be simply assembled to complete the present micromixer. The number of the modified Tesla units was varied from three to five. The tip clearance between the divider structure and the micromixer wall is expected to play a key role in generating flow disturbance and is varied in the range from 40 μm to 80 μm. It is to 20~40% of the present micromixer depth. The mixing performance was simulated in terms of the degree of mixing (DOM) at the outlet and the required pressure load between the inlets and outlet and compared with those of the modified planar Tesla micromixer without tip clearance.

A numerical approach has several benefits, such as easy visualization of the mixing process and the associated flow patterns. Accordingly, it is widely used in studying the mixing performance of a micromixer. For a numerical study, a commercial software is commonly used. For example, Makhsuda et al. [19] used the commercial software ANSYS^®^ Fluent [27] to study the mixing performance. Rhoades et al. [28] used the commercial software COMSOL Multiphysics 5.1 (COMSOL, Inc., Burlington, MA, USA) to simulate the mixing performance of a grooved serpentine micro-channel. Volpe et al. [29] used the lattice Boltzmann method (LBM) to study the flow dynamics of a continuous size-based sorter microfluidic device. In this paper, the mixing performance of the present micromixer was simulated using the commercial software ANSYS^®^ Fluent 2021 R2 [27].

Most micromixers for biological and chemical applications operate in the range of millisecond mixing time, and the corresponding Reynolds number is less than about 100 [30,31,32]. In this range of the Reynolds number, the micromixing is governed by two distinct mechanisms such as the molecular diffusion and convection [18,22]. The Reynolds number is usually categorized into three regimes according to the dominant mixing mechanism: molecular dominance, transition, and convection dominance. The present numerical study was carried out to cover all of the three mixing regimes. Therefore, the Reynolds number was varied from 0.1 to 80, and the corresponding volume flow rate ranged from 1.3 μL/min to 964.6 μL/min.

## 2. Modified Tesla Micromixer with Tip Clearance

Figure 1 shows a modified Tesla mixing unit. It is placed serially in the present passive micromixer. The wedge-shape divider splits the fluid stream into two sub-streams which recombine downstream: sub-stream 1 and sub-stream 2. Therefore, the mixing performance of each modified Tesla mixing unit shows some dependence on the geometry of the divider. The detailed geometry of the modified Tesla mixing unit is the same as one optimized by Hossain et al. [17]. In Figure 1b, h is the height of tip clearance, and it is varied from 40 μm to 80 μm. According to previous research [17,23], multiple vortices form at the vortex zone as the Reynolds number increases; the outlet of each Tesla mixing unit is named as the vortex zone.

Figure 2 shows a schematic diagram of the present passive micromixer. The cross section of the inlet and outlet branches is rectangular: 200 μm wide and 200 μm deep. Both inlets 1 and 2 are 1000 μm long while the outlet branch is 500 μm long. Even Figure 2 shows four modified Tesla mixing units, and the actual number of the modified Tesla mixing units is varied from three to five. The Ss in Figure 2 indicate the cross section at the outlet of each modified Tesla mixing unit. For example, S4 is the cross section after the fourth Tesla mixing unit. As the two inlets are facing opposite to each other, micromixing takes place mainly in the modified Tesla mixing units.

## 3. Governing Equations and Computational Procedure

As the fluid was assumed Newtonian and incompressible, the following continuity and Navier–Stokes equations are the governing equations:(1)$$\left(\vec{u}\cdot \nabla \right)\vec{u}=-\frac{1}{\rho }\nabla p+\nu \nabla^{2}\vec{u}$$
(2)$$\nabla \cdot \vec{u}=0$$
where $\vec{u}$, *p*, and *ν* are the velocity vector, pressure, and kinematic viscosity, respectively. The evolution of mixing was simulated by solving an advection-diffusion equation:(3)$$\left(\vec{u}\cdot \nabla \right)\varphi =D\nabla^{2}\varphi$$
where *D* and *φ* are the mass diffusivity and mass fraction of fluid A, respectively.

ANSYS^®^ FLUENT 2021 R2, Canonsburg, PA, USA [25] was used to solve the governing Equations (1)–(3). It is based on the finite volume method. The QUICK scheme (quadratic upstream interpolation for convective kinematics) was used to discretize the convective terms in Equations (1) and (3), and its theoretical accuracy is third order. The velocity distribution at the two inlets was assumed as uniform, and the outflow condition was used at the outlet. The no-slip boundary condition was specified along the all walls were treated as a no-slip boundary. The mass fraction of fluid A is *φ* = 1 at inlet 1 and *φ* = 0 at inlet 2.

The mixing performance of a combined micromixer was evaluated using the degree of mixing (DOM) and mixing energy cost (MEC). The DOM is defined in the following form:(4)$$DOM=1-\frac{1}{\xi }\sqrt{\overset{n}{\underset{i=1}{\sum }}\frac{{\left(\varphi_{i}-\xi \right)}^{2}}{n}},$$
where *φ_i_* and *n* are the mass fraction of fluid A in the *i*th cell and the total number of cells, respectively; *ξ* = 0.5, which means equal mixing of the two fluids. The MEC is widely used to evaluate the effectiveness of the present micromixer and is defined by combining the pressure load and DOM in the following form [33,34]:(5)MEC=∆pρumean2DOM×100,
where $u_{mean}$ is the average velocity at the outlet, and ∆p is the pressure load between the inlet and the outlet.

The aqueous fluids flowing into the two inlets were assumed to have the same properties, the same as the physical properties of the water. Therefore, the density, diffusion constant, and viscosity of the fluid are *ρ* = 997 kg/m^3^, *D* = 1.0 × 10^−10^ m^2^ s^−1^, and *ν =* 0.89 × 10^−6^ m^2^ s^−1^, respectively. The corresponding Schmidt (Sc) number is approximately 10^4^ (the ratio of the kinetic viscosity and the mass diffusivity of the fluid). The Reynolds number was defined as Re=ρUmeandhμ, where $\rho ,\;\;U_{mean},\;\;d_{h},\;\mathrm{and}\;\mu$ mean the density, the mean velocity at the outlet, the hydraulic diameter of the outlet channel, and the dynamic viscosity of the fluid, respectively.

## 4. Validation of the Numerical Study

Accurate numerical simulation is still a challenging problem to study the mixing in micromixers, especially for high Sc numbers. Many research papers do not deal with this computational issue. In general, the numerical diffusion can deteriorate the accuracy of the simulated results for high Sc number simulations. To obtain a quantitatively more rigorous numerical solution, we could use either a particle-based simulation method such as Monte Carlo method [35] or decrease the cell Peclet number for a grid-based method. Here, the cell Peclet number is defined as $Pe=\frac{U_{cell}l_{cell}}{D}$, where $U_{cell}$ and $l_{cell}$ are the local flow velocity and cell size, respectively. However, these methods are computationally expensive to adopt in a study such as this paper. As a practical remedy, most numerical studies prefer a detailed study of grid independence by comparing with experimental data [15,36].

The present numerical approach was validated by simulating the micromixer examined by Chung et al. [24]. Figure 3 shows a schematic diagram of the micromixer, and it consists of three mixing units. Each mixing unit contains three rectangular baffles and the associated gaps; the thickness of the baffles is 80 μm. As each baffle is shorter than the micromixer width, a gap is created. The first two baffles form a gap in the center while the third baffle makes two gaps around its edges as shown in Figure 3. The width of the inlet 1 and two side inlets, inlet 2 and inlet 3, are 400 μm and 200 μm, respectively. The depth of the micromixer is 130 μm.

The density, diffusion constant and viscosity of the fluid are *ρ* = 997 kg/m^3^, *D* = 3.6 × 10^−10^ m^2^ s^−1^, and *ν* = 0.89 × 10^−6^ m^2^ s^−1^, respectively. Therefore, the Schmidt (Sc) number is approximately 40,000. The simulation was carried out and compared with the corresponding experimental data for Reynolds numbers Re = 60. Here, the Reynolds number is defined as Re=ρUmeandhμ*,* where $\rho ,\;\;U_{mean},\;\;d_{h},\;\mathrm{and}\;\mu$ indicate the density, the mean velocity at the outlet, the hydraulic diameter of the outlet channel ($d_{h}=196.2\;\mu m$), and the dynamic viscosity of the fluid, respectively. Structured hexahedral cells were used to mesh the computational domain; the total number of cells is about 3.75 million.

Figure 4 compares the present simulation images with the corresponding experimental data reported by Chung et al. [24]. The mixing images at the two different depth show that the mixing process is quite depth-dependent: a strong mixing in the cross-flow direction. The comparison confirms that the present numerical simulation captures all the important mixing features such as the formation of vortices around short baffles.

Prior to the present numerical study, an additional set of preliminary simulations was carried out to determine an appropriate cell size for the present micromixer. For this study, the edge size of cells was varied from 4.5 μm to 6 μm for three modified Tesla units. The corresponding number of mesh varies from 1.8 × 10^6^ to 3.8 × 10^6^. The simulation was carried out for Re = 0.5. Figure 5 shows the dependence of the calculated DOM on the edge size. The deviation of 5 μm solution from that of 4.5 μm is about 1%. Therefore, 5 μm is small enough to obtain grid independent solutions.

Using the numerical solutions, the grid convergence index (GCI) was also calculated to quantify the uncertainty of grid convergence [37,38]. According to the Richardson extrapolation methodology, the GCI is calculated as follows:(6)$$GCI=F_{s}\frac{\left|\varepsilon \right|}{r^{p}-1},$$
(7)$$\varepsilon =\frac{f_{coarse}-f_{fine}}{f_{fine}},$$
where *F_s_*, *r*, and *p* are the safety factor of the method, grid refinement ratio, and the order of accuracy of the numerical method, respectively. *f_coarse_* and *f_fine_* are the numerical results obtained with a coarse grid and fine grid, respectively. *F_s_* was specified at 1.25 as suggested by Roache [37]. For the edge size of 4.5 μm, 5 μm, and 6 μm, the corresponding number of nodes are 3.8 × 10^6^, 2.98 × 10^6^, and 1.8 × 10^6^ for three Tesla mixing units, respectively; and 5.9 × 10^6^, 4.4 × 10^6^, and 2.6 × 10^6^ for five mixing units, respectively. As a result, the GCI of the computed DOM is reduced from 5.7% to 1.1%. Therefore, the edge size of 5 μm was chosen to obtain the present numerical solutions.

According to Okuducu et al. [39], the accuracy of numerical solutions is also dependent on the type of cells. Structured hexahedral cells show the most reliable numerical solution, in comparison with tetrahedral and prism cells. In this paper, most cells were generated to be hexahedral as can be seen in Figure 5. The number of prism cells was minimized; refer to the red circle in Figure 5.

## 5. Results and Discussion

The present micromixer with tip clearance was simulated to assess its mixing performance by comparing with that of the Tesla micromixer without any tip clearance for Reynolds numbers from 0.1 to 80. The velocity at the two inlets was specified as uniform in the range from 0.2512 mm/s to 200.96 mm/s. Therefore, the corresponding volume flow rates range from 1.2 μL/min to 964.6 μL/min. The mixing performance was evaluated in terms of the DOM at outlet and the corresponding MEC.

Figure 6 shows the DOM of the modified Tesla micromixer with tip clearance h = 60 μm against that of no tip clearance; N indicates the number of the modified Tesla mixing units. A noticeable enhancement of the DOM is observed in the range of the Reynolds numbers Re < 50, and the amount of improvement increases with the number of the modified Tesla mixing units. For the Reynolds number of Re = 20 and N = 4, the DOM with tip clearance h = 60 μm is 24% higher than that with no tip clearance. When the Reynolds number is larger than about 50, the tip clearance allows a more efficient operation in terms of the required pressure load. For example, the DOM of tip clearance h = 40 μm for Re = 80 and N = 5 shows almost the value with the case of no tip clearance while it reduces the pressure load by about 8%.

Figure 7 shows the mixing performance map in terms of the DOM versus the required pressure load for four Reynolds numbers, Re = 0.1, 1, 10, and 50. The dotted line in the figure indicates the variation of the DOM for the case with no tip clearance. The DOM shows a linear relationship with the number of the mixing units. It means that the DOM can be improved linearly at the expense of the pressure load between the inlets and outlet, increasing the number of mixing units. On the other hand, for the case of tip clearance, all the DOM except for Re = 50 show an additional increment of the DOM from that of the modified Tesla micromixer. For example, the DOM of tip clearance h = 60 μm is 94% higher than that of the case of no tip clearance for Re = 5 and N = 4. At the same time, the required pressure load is 12% reduced. It is also noteworthy that the height of tip clearance is optimized in terms of the DOM. The optimum value of h is roughly h = 60 μm for most cases, and it is about 33% of the present micromixer depth. As the Reynolds number is increased to larger than about 50, the effects of tip clearance become less significant. It is associated with twis iso symmetric counter-rotating vortices formed at the outlet branch of each modified Tesla unit; refer to the vortex zone in Figure 1a. This flow characteristic is also reported by Hossain et al. [17] and is described in detail later.

In the low Reynolds number regime of Re ≤ 1, the molecular diffusion dominates mixing process so that a straight channel is a good reference to compare with. Figure 8 compares the mixing evolution of the present micromixer with that of a straight channel; the DOM was obtained just after each Tesla mixing unit. The micromixers based on the modified Tesla mixing units show a noticeable mixing enhancement from the straight channel, and the enhancement increases with the number of mixing units. For the Reynolds number of Re = 0.1, the mixing enhancement of the modified Tesla mixing unit after one mixing unit is 63% and increases to 75% after five mixing units. The tip clearance results in an additional enhancement. It is 13% after one mixing unit and increases to 25% after five mixing units. A similar enhancement of the DOM is observed for the Reynolds number of Re = 1.

Figure 9 shows the mixing effectiveness of the present micromixer in comparison with that of the modified Tesla micromixer. In the figure, a smaller value of the MEC means more effective and requires a lesser pressure load to obtain the same degree of mixing. Therefore, the tip clearance of the present micromixer is found to reduce significantly the required pressure load for all Reynolds numbers, even though the effect is quite limited for the Reynolds number of Re = 50. Another interesting thing is that the tip clearance is optimized to minimize the MEC. The optimum value is about 60 μm, and close to the value for maximizing the DOM shown in Figure 6. This suggests that the size of tip clearance can be determined to enhance the DOM as well as minimize the required pressure load.

Figure 10 shows the increment of the DOM obtained by embedding tip clearance into the modified Tesla mixing units. The vertical axis indicates the increment of the DOM obtained with tip clearance from the DOM with tip clearance. Therefore, a negative value means that the tip clearance affects the DOM in a negative way. For Re = 0.1, the effects of tip clearance is significant throughout the whole mixing unit; in the molecular dominance regime of mixing. On the contrary, the tip clearance in the first mixing unit takes little or negative effects on mixing for Re = 1, 5 and 10; in the intermediate range of the Reynolds number. The increment of the DOM increases as it goes downstream. This suggests that the flow characteristics associated with the mixing enhancement in the two ranges are different from each other.

The mixing enhancement mechanism is analyzed further in the three different regimes of mixing: molecular diffusion dominance, transition, and convection dominance. Figure 11 shows the concentration contours on two planes for the Reynolds number of Re = 0.1: at z = 30 μm and 100 μm. Here, the plane at z = 100 μm corresponds to the mid-depth plane of the present micromixer while the plane at z = 30 μm is in the middle of the tip clearance. The case of tip clearance h = 60 μm shows a wider interface between the two fluids on the plane at z = 30 μm; this means that the mixing along the interface is more active. Figure 12 plots the concentration and the velocity vector on the yz plane for Re = 0.1 and h = 60 μm. The flow through tip clearance drags the interface of the sub-stream 1 and connects it to the interface of the sub-stream 2: drag and connection of the interface by tip clearance flow. This kind of flow characteristic is seen both on section 1 and 2 while the two interfaces for the case of no tip clearance is separated by the structure, as seen in Figure 11b. This explains why the increment of the DOM for Re = 1 is significant throughout the whole mixing unit. Therefore, the drag and connection of interface is the main flow mechanism for the mixing enhancement caused by the tip clearance in the molecular diffusion regime of mixing.

As the Reynolds number increases, the convective mixing becomes significant, and the mixing enhancement by tip clearance is obtained in a different way. Figure 13 compares the concentration contours on the xy planes for the Reynolds number of Re = 5. For the case of tip clearance, the interface between the two fluids appears wavier and multiple times and is a result of the mixing enhancement. The difference caused by tip clearance is more obvious on the plane at z = 100 μm. Figure 14 compares the concentration contours with the corresponding velocity vector on the two yz planes for Re = 5: section 1 and 2. For the case of no tip clearance, a vortex forms on the cross-section 1, and develops into a pair of two counter-rotating vortices on the cross-section 2.

Contrarily, the vortex flow on section 1 is agitated and developed in a different way on section 2 for the case of tip clearance. This different flow evolution suggests that the flow passing through tip clearance interacts with the secondary flow generated on section 1 and develops into an agitated vortex flow on section 2. The agitated vortex flow plays a significant role in the mixing enhancement for the Reynolds numbers 1 ≤ Re < 50. Figure 15 compares the concentration contours with the corresponding velocity vector on the two yz planes for Re = 10: section 1 and 2. For the case of tip clearance, the flow through tip clearance agitates the secondary flow generated in the vortex zone and confirms the agitated vortex flow to cause the mixing enhancement.

As the Reynolds number increases further, the vortex flow formed in the vortex zone of each Tesla mixing unit develops into a pair of strong counter-rotating vortices. They are separated distinctly and become stronger. Figure 16 compares the concentration contours on the two xy planes for the Reynolds number of Re = 50. Comparing the concentration contours on the plane at z = 100 μm, the mixing seems to be processed in a similar way, even though there is a little difference locally. On the contrary, the concentration contours on the plane at z = 30 μm show a more vivid difference between them. The case of tip clearance shows a more complicated pattern of the interface. This is caused by the asymmetric geometry due to tip clearance. This flow pattern is observed on both cross sections of section 1 and 2. This difference of interface pattern suggests that the influence of tip clearance is localized, and there is another significant mechanism of mixing. Figure 17 compares the concentration contours with velocity vector on the two yz planes. It shows that there are two distinct counter-rotating vortices on both planes of section 1 and 2, irrespective of tip clearance. They seem almost symmetric even for the case of tip clearance. This suggests that the mixing is mainly governed by the two counter-rotating vortices for the Reynolds numbers Re ≥ 50; this pair of counter-rotating vortices was also reported in the previous study [17,21].

Figure 18 shows how the presence of tip clearance affects the vortex patterns and mixing performance in the present micromixer for the Reynolds of Re = 80. At the cross section 1 and 2, a pair of vortices seems formed, and this is generated as the flow follows the circular passage of the micromixer, at the first and last Tesla mixing unit. It implies that the centrifugal force plays a significant role. Another interesting thing is that the vortex pattern is little affected as long as the tip clearance remains about h ≤ 60 μm. On the other hand, the vortex close to the tip clearance zone (lower vortex) was observed to have noticeably shrunk for the tip clearance h = 70 μm, as can be seen in Figure 18a. This suggests that the convection flow is strong enough to localize the effects of tip clearance as long as the tip clearance is smaller than about h = 70 μm. Another pair of vortices are seen at section 3 and 4, which are located around the outlet of the first and last Tesla mixing units.

Accordingly, the flow characteristics such as a pair of vortices play a role as a governing mechanism in the convection-dominant regime of mixing. As a result, the concentration on plane 4 for no tip clearance and h = 60 μm is almost identical, as can be seen in Figure 18b. Therefore, the presence of tip clearance contributes a little to the mixing enhancement for the Reynolds numbers Re ≥ 50 as long as the tip clearance remains smaller than about h = 70 μm.

Figure 19 shows how the mixing evolves throughout the mixing units for Re = 50. Unlike the case of no tip clearance in Figure 19a, the case of tip clearance in Figure 19b shows asymmetric concentration contours on the upper section of yz planes of section 1 and 2; the upper section corresponds to the sub-stream 1. However, symmetry seems to be recovered mostly on the lower section of the yz planes; refer to the box of red dotted lines in the figure. The lower section corresponds to the yz plane in sub-stream 2. This recovery is attributed to the two symmetric counter-rotating vortices depicted in Figure 15 and suggests that the effects of tip clearance are localized for Re = 50. The two counter-rotating vortices generated in the vortex zone are strong enough to recover the asymmetric mixing pattern in the tip clearance zone. No significant increment of the DOM is achieved by the tip clearance for Re ≥ 50, as shown in Figure 6.

## 6. Conclusions

This paper studied numerically the effects of tip clearance on the mixing performance of the modified Tesla micromixer. The present micromixer consists of several modified Tesla mixing units, and each mixing unit has tip clearance above the wedge-shape divider. The numerical simulation was carried out for the Reynolds numbers 0.1 ≤ Re ≤ 80 and three different numbers of mixing units: 3, 4 and 5. The mixing performance was assessed in terms of the DOM at the outlet and the required pressure load between the inlets and outlet. The mixing performance was simulated using the commercial software ANSYS^®^ Fluent 2021 R2.

The effects of tip clearance were found noticeably over a wide range of the Reynolds numbers, Re < 50. For example, the DOM of tip clearance h = 60 μm is 94% higher than that with no tip clearance for Re = 5 and N = 4, and in addition, the required pressure load is 12% reduced. The height of tip clearance is optimized in terms of the DOM, and the optimum value is roughly h = 60 μm for most cases. It corresponds to 33% of the present micromixer depth. The tip clearance is also optimized to minimize the MEC. The optimum value is close to that for maximizing the DOM. The size of tip clearance can be determined to enhance the DOM as well as minimize the required pressure load.

The mixing enhancement due to tip clearance was obtained by different mixing mechanisms in accordance with the Reynolds number. In the molecular diffusion regime of mixing, Re ≤ 1, the mixing enhancement is obtained mainly by connection of the two interfaces in sub-stream 1 and sub-stream 2. The flow through the tip clearance drags the interface in sub-stream 1 and connects it to the interface in sub-stream 2. This flow characteristic causes the mixing to happen actively and the interface to become wider in the tip clearance zone.

The mixing enhancement in the intermediate range of the Reynolds number, 1 < Re ≤ 50, is attributed to the interaction of the flow through tip clearance and the secondary flow in the vortex zone of each Tesla mixing unit. Unlike for the case of no tip clearance, the flow through tip clearance agitates the secondary flow formed in the vortex zone of each Tesla mixing unit, and it leads to an increment in the DOM.

When the Reynolds number is larger than about 50, vortices are formed at various locations and drive the mixing in the modified Tesla micromixer. For the Reynolds number of Re = 80, a pair of vortices is formed around the inlet and outlet of each Tesla mixing unit. This vortex pattern is little affected by the presence of tip clearance as long as the tip clearance remains smaller than about h = 70 μm. It plays a role as a governing mechanism for the present micromixer in the convection-dominant regime of mixing. As a result, the DOM at the outlet is little enhanced by the presence of tip clearance. The tip clearance contributes only to reduce the required pressure load for the same value of the DOM.

The tip clearance embedded into the modified Tesla micromixer was shown to improve the mixing performance over a wide range of the Reynolds numbers. The improvement of mixing performance is achieved in terms of the DOM enhancement as well as the reduction of the corresponding pressure load. The mixing enhancement mechanism is dependent on the magnitude of the Reynolds number. The tip clearance is easily realized by lowering the height of the wedge-shape divider of the modified Tesla micromixer.

## Figures and Tables

**Figure 1 micromachines-13-01375-f001:**
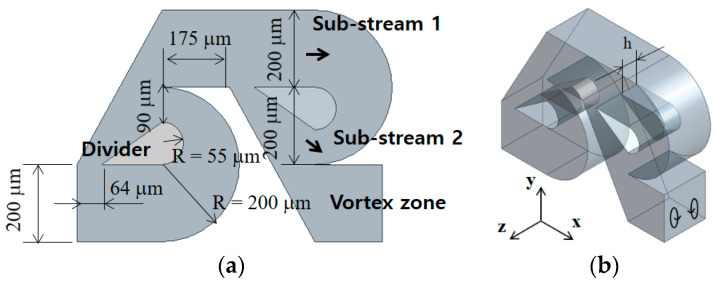
Schematic diagram of the modified Tesla mixing unit (non-proportional): (**a**) Front view and (**b**) Three-dimensional view.

**Figure 2 micromachines-13-01375-f002:**
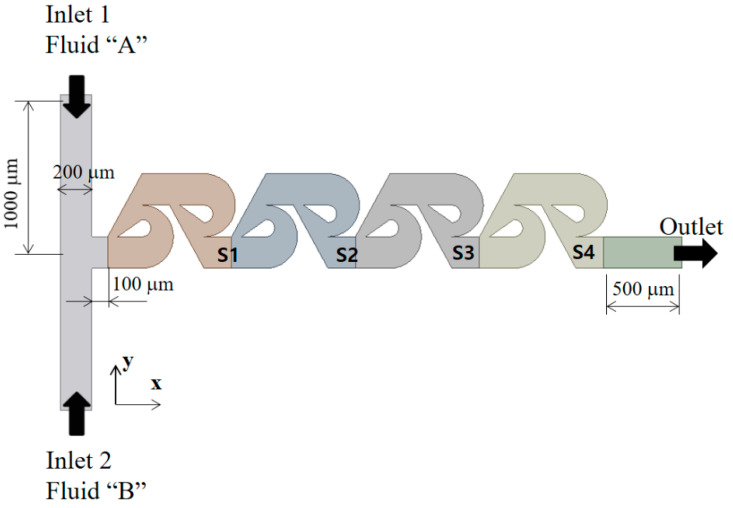
Schematic diagram of present micromixer (non-proportional).

**Figure 3 micromachines-13-01375-f003:**
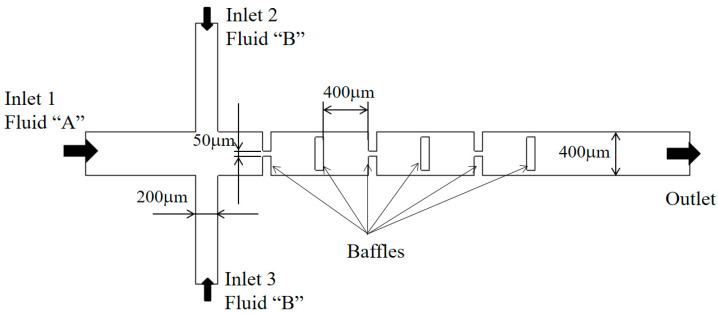
Diagram of the micromixer experimented by Chung et al. [24].

**Figure 4 micromachines-13-01375-f004:**
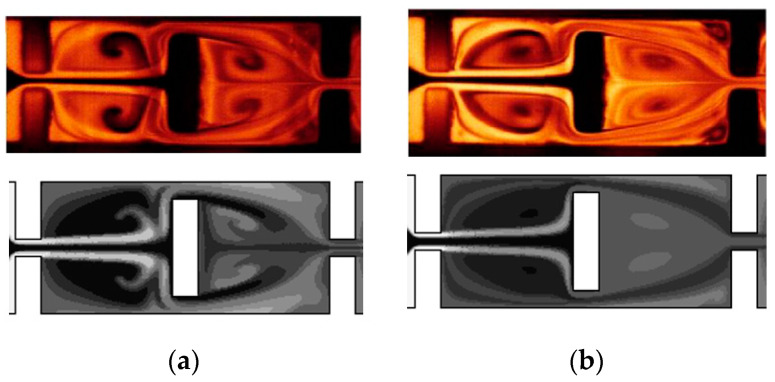
Comparison of the mixing images of the first two chambers for Re = 60: (**a**) At z = 32.5 μm and (**b**) At z = 65 μm.

**Figure 5 micromachines-13-01375-f005:**
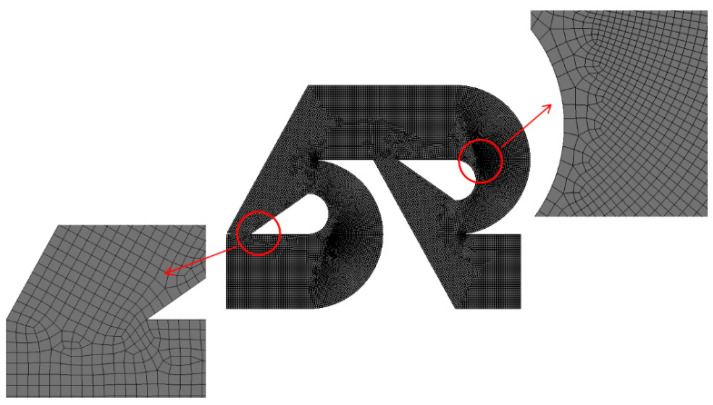
Grid for a modified Tesla mixing unit.

**Figure 6 micromachines-13-01375-f006:**
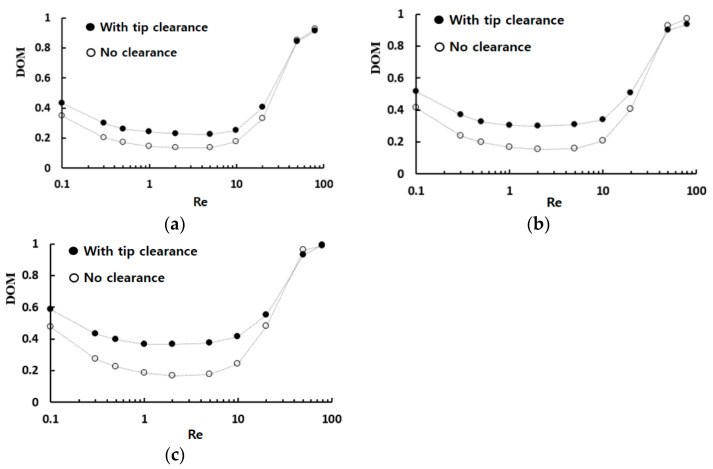
Enhancement of the DOM by tip clearance: (**a**) N = 3, (**b**) N = 4, and (**c**) N = 5.

**Figure 7 micromachines-13-01375-f007:**
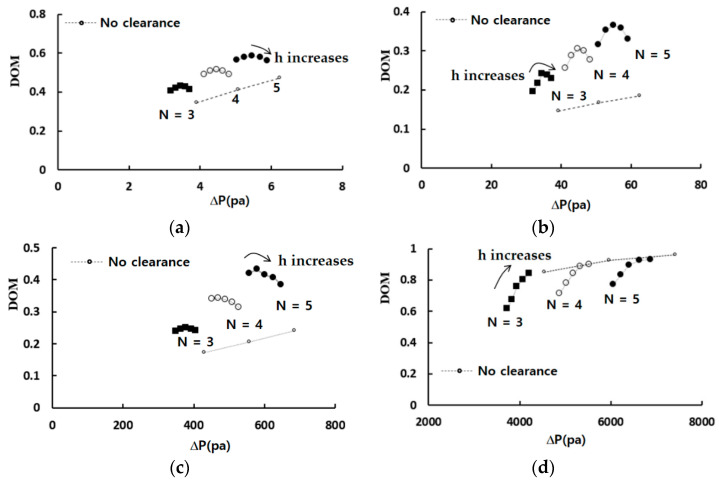
Mixing performance map: (**a**) Re = 0.1, (**b**) Re = 1, (**c**) Re = 10, and (**d**) Re = 50.

**Figure 8 micromachines-13-01375-f008:**
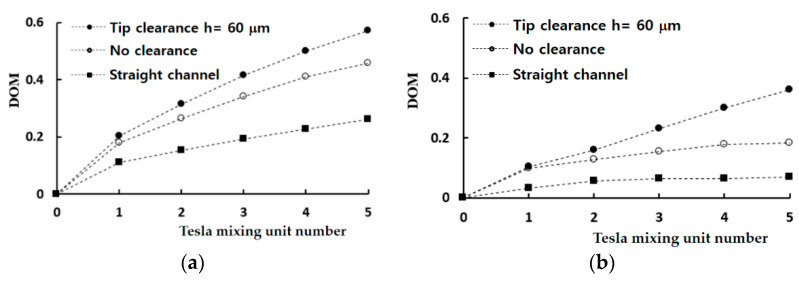
Comparison of mixing evolution in the axial direction: (**a**) Re = 0.1 and (**b**) Re = 1.

**Figure 9 micromachines-13-01375-f009:**
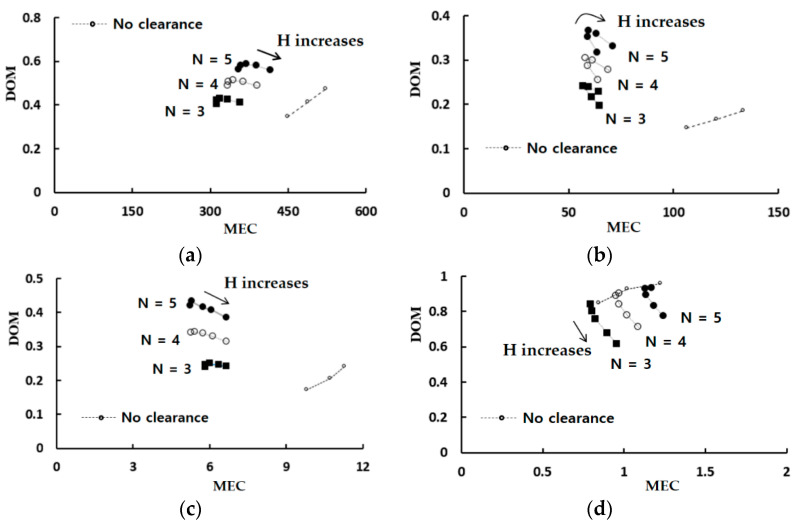
Mixing effectiveness map: (**a**) Re = 0.1, (**b**) Re = 1, (**c**) Re = 10, and (**d**) Re = 50.

**Figure 10 micromachines-13-01375-f010:**
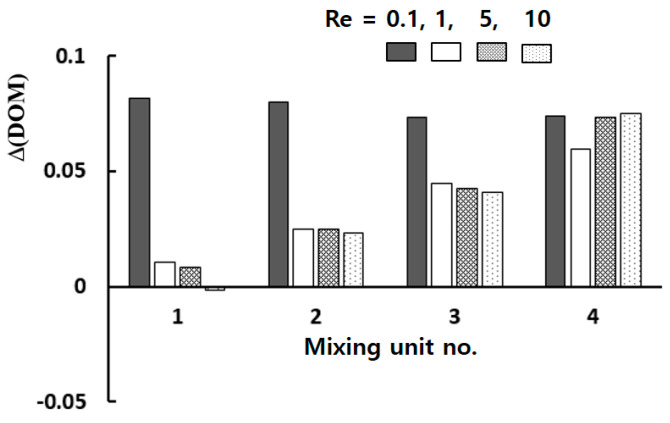
Increment of the DOM by tip clearance.

**Figure 11 micromachines-13-01375-f011:**
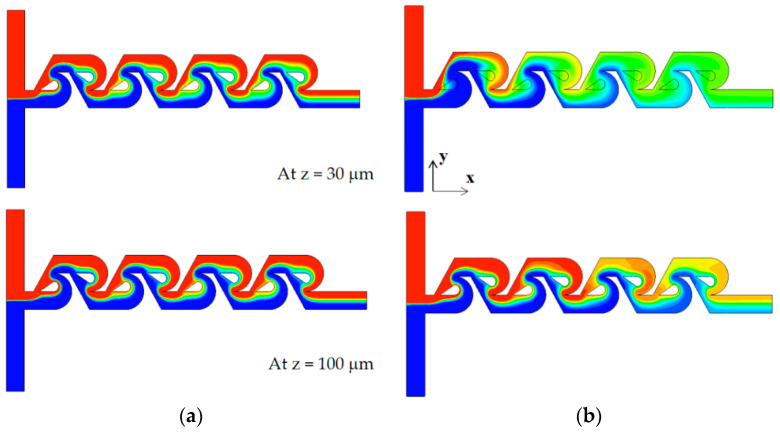
Comparison of concentration contours on the xy planes for Re = 0.1: (**a**) no clearance and (**b**) with tip clearance h = 60 μm.

**Figure 12 micromachines-13-01375-f012:**
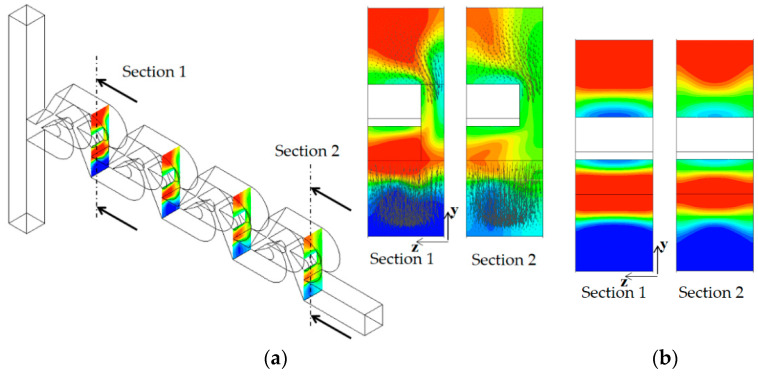
Evolution of mixing and velocity vector on the yz planes for Re = 0.1: (**a**) tip clearance h = 60 μm and (**b**) no tip clearance.

**Figure 13 micromachines-13-01375-f013:**
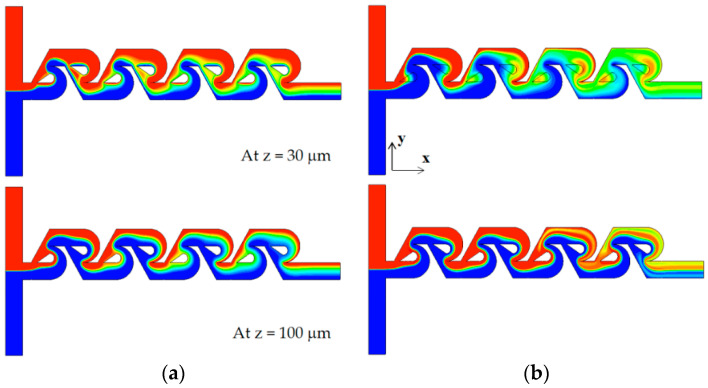
Comparison of concentration contours on the xy planes for Re = 5: (**a**) no clearance and (**b**) with tip clearance h = 60 μm.

**Figure 14 micromachines-13-01375-f014:**
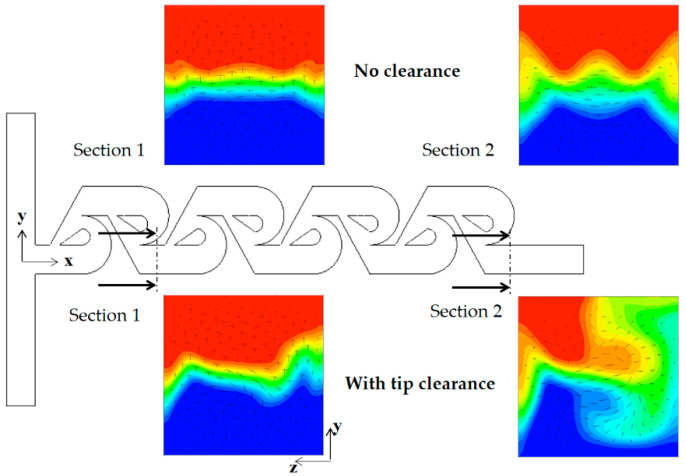
Concentration contours with velocity vector on the yz planes for Re =5.

**Figure 15 micromachines-13-01375-f015:**
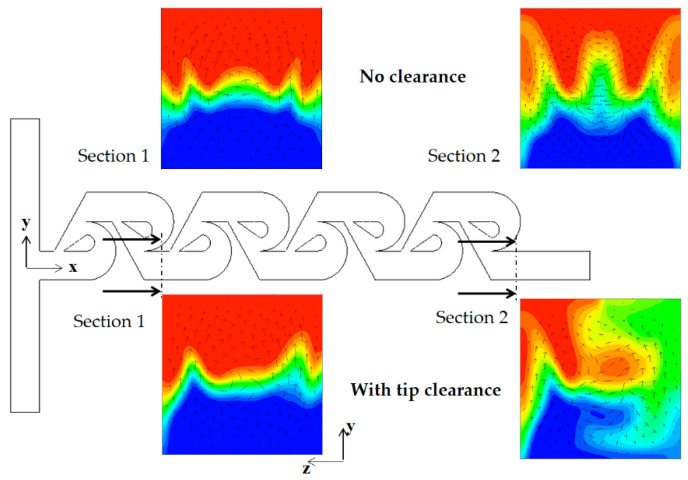
Concentration contours with velocity vector on the yz planes for Re = 10.

**Figure 16 micromachines-13-01375-f016:**
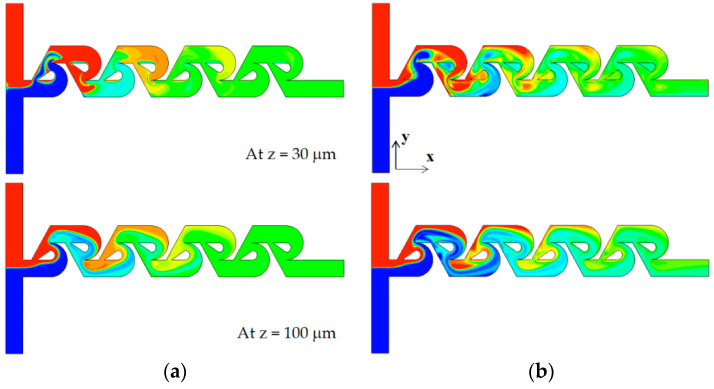
Comparison of concentration contours on the xy planes for Re = 50: (**a**) no clearance and (**b**) with tip clearance h = 60 μm.

**Figure 17 micromachines-13-01375-f017:**
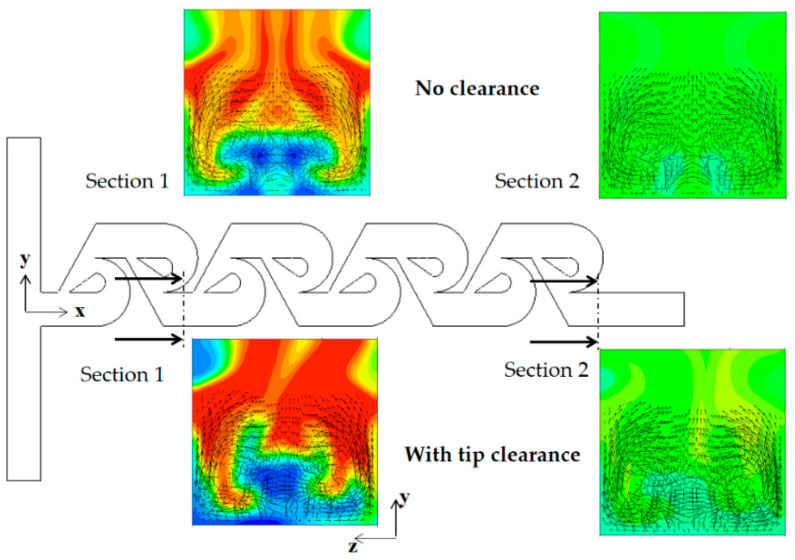
Concentration contours and velocity vector on the yz planes for Re = 50.

**Figure 18 micromachines-13-01375-f018:**
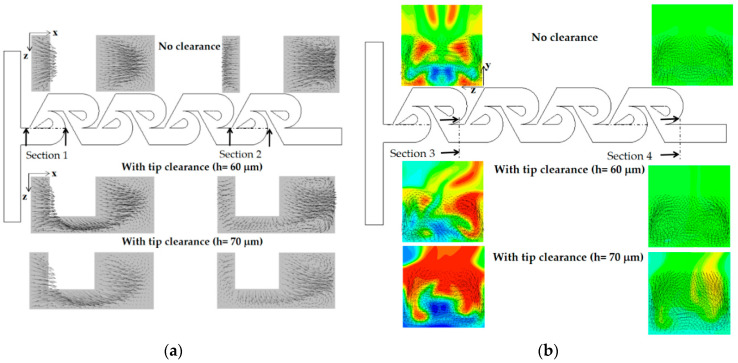
Concentration contours and velocity vector on the several planes for Re = 80: (**a**) velocity vector (**b**) concentration contours.

**Figure 19 micromachines-13-01375-f019:**
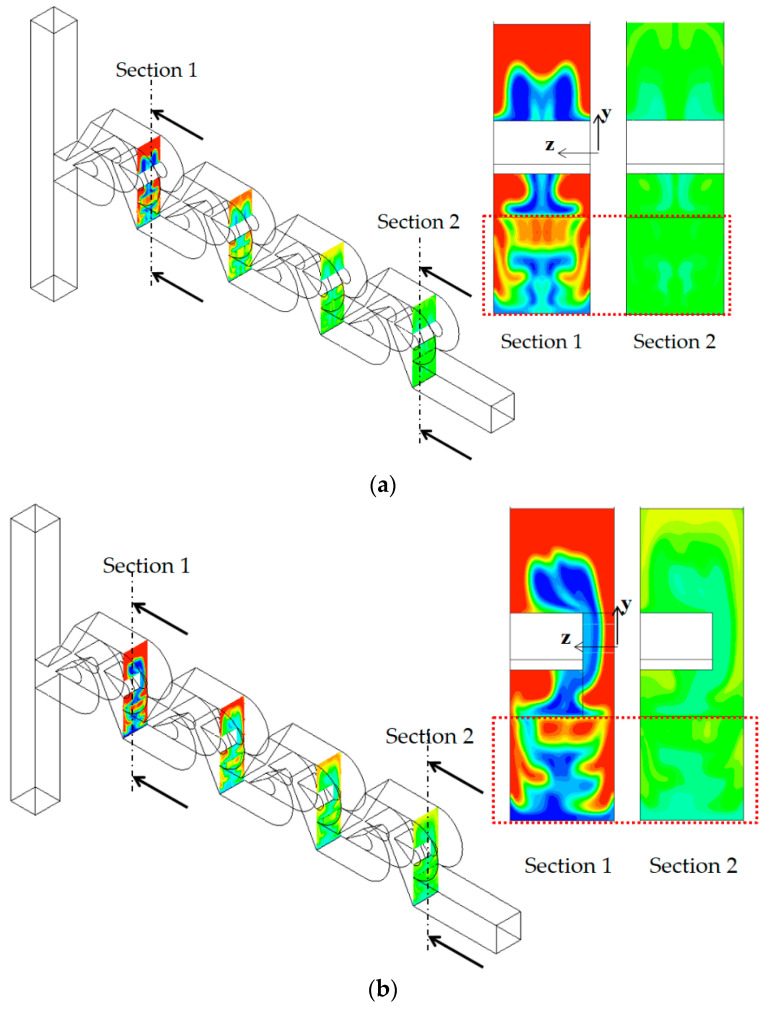
Evolution of mixing on the yz planes for Re = 50: (**a**) no tip clearance and (**b**) tip clearance h = 60 μm.

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
