# Peer review of "Mixing Performance of the Modified Tesla Micromixer with Tip Clearance"

_micromachines, 2022, doi:10.3390/mi13091375_

Round 1

Reviewer 1 Report

The authors have described a passive TESLA micromixer here. They have analyzed the degree of mixing and the mixing performance well. However, a few important aspects remain that need to be taken care of before this manuscript can be accepted.

The introduction needs some more details on how the proposed micromixer can be actually made. One of the ways could be Xurography. Please see the paper below that can be used:

https://doi.org/10.1016/j.cherd.2021.01.022

As it is a theoretical paper, a comparison of mixing with respect to a straight channel with the same channel length needs to be shown. Please see the above paper for a theoretical equation.

 A discussion on the mixing cost must be shown.

Author Response

Reviewer #1

Thanks for reviewing the paper.

1) The authors have described a passive TESLA micromixer here. They have analyzed the degree of mixing and the mixing performance well. However, a few important aspects remain that need to be taken care of before this manuscript can be accepted. The introduction needs some more details on how the proposed micromixer can be actually made. One of the ways could be Xurography. Please see the paper below that can be used:

https://doi.org/10.1016/j.cherd.2021.01.022

Answer> As you suggested, the manuscript was revised (lines 97~102):

As the structure of present micromixer is slightly modified from that of a planar micromixer, microfabrication techniques such as Xurography [25] can be easily applied. The Xurography technique uses thin, pressure sensitive double-sided adhesive flexible films so that the tip clearance zone is simply tailored using a cutter plotter; it cuts off a film along the perimeter of the modified Tesla micromixer. The tailored film and the planar structure can be simply assembled to complete the present micromixer.

2) As it is a theoretical paper, a comparison of mixing with respect to a straight channel with the same channel length needs to be shown. Please see the above paper for a theoretical equation.

Answer> As you suggested, the manuscript was revised (lines 353~363):

As you know well, the analytic solution is based on the velocity profile derived under the lubrication approximation, and valid only for Re << 1. The cross section of the present micromixer is height/width = 1. So, the numerical simulation results of a straight channel was used to compare against that of the present micromixer. As the straight channel shows a poor performance in the convection dominant regime of mixing, the comparison was made only in the molecular diffusion dominant regime of mixing; for Re = 0.1 and 1.

In the low Reynolds number regime of Re  1, the molecular diffusion dominates mixing process so that a straight channel is a good reference to compare with. Figure 7 compares the mixing evolution of the present micromixer with that of a straight channel; the DOM was obtained just after each Tesla mixing unit. The micromixers based on the modified Tesla mixing units show a noticeable mixing enhancement from the straight channel, and the enhancement increases with the number of mixing units. For the Reynolds number of Re = 0.1, the mixing enhancement of the modified Tesla mixing unit after one mixing unit is 63 %, and increases to 75 % after five mixing units. The tip clearance results in an additional enhancement. It is 13 % after one mixing unit, and increases to 25 % after five mixing units. A similar enhancement of the DOM is observed for the Reynolds number of Re = 1.

3) A discussion on the mixing cost must be shown.

Answer> As you suggested, the manuscript was revised (lines 189~193, 364~373, and Figure 8):

The MEC is widely used to evaluate the effectiveness of the present micromixer and is defined by combining the pressure load and DOM in the following form [31, 32]:

                 (5)

where  is the average velocity at the outlet, and  is the pressure load between the inlet and the outlet.

Figure 8 shows the mixing effectiveness of the present micromixer in comparison with that of the modified Tesla micromixer. In the figure, a smaller value of the MEC means more effective and requires a lesser pressure load to obtain the same degree of mixing. Therefore, the tip clearance of the present micromixer is found to reduce significantly the require pressure load for the all Reynolds numbers; even though the effect is quite limited for the Reynolds number of Re = 50. Another interesting thing is that the tip clearance is optimized to minimize the MEC. The optimum value is about 60 mm, and close to the value for maximizing the DOM shown in Figure 6. This suggests that the size of tip clearance can be determined to enhance the DOM as well as minimize the required pressure load.

Figure 8. Mixing effectiveness map: (a) Re = 0.1, (b) Re = 1, (c) Re = 10, and (d) Re = 50.

Reviewer 2 Report

The authors presented an interesting research on the effects of tip clearance on the mixing performance of the modified Tesla micromixer. The manuscript is reasonable in structure and detailed in content. However, some problems still exist. It is advised to accept them after modification.

1. The first letter of the word “modified” in the paper title should be capitalized.

2. Two section titles are numbered “4”, the authors should adjust them.

3. The unit should be added for the number “65” in the demonstration of Figure 4.

4. Not all of the sub-images in Figure 5 are numbered, and descriptions of all images are missing.

5. Description of sub-image (b) is missing in Figure 8 and 10, 13.

6. ANSYS Fluent 2021 R2 is mentioned in the line 108 and 636, but ANSYS FLUENT 2020 R2 is described in the line 168. The authors need to reconfirm the version of the ANSYS Fluent software.

7. The authors should add more detailed demonstrations and simulation data to explain why tip clearance has little effects on the mixing performance when Re >50.

8. In the “Abstract” and “Conclusion” part, the authors show the simulation work is conducted over a wide range of the Reynolds numbers from 0.1 to 80. The authors should add simulation data and result discussion about Re around 80 in the paper content.

9. The lateral structure mixer may be related this mixer . The author may refer: An Enhanced One-Layer Passive Microfluidic Mixer With an Optimized Lateral Structure With the Dean Effect, J. Fluids Eng. Sep 2015, 137(9): 091102.

Author Response

Reviewer #2

Thanks for reviewing the paper.

The authors presented an interesting research on the effects of tip clearance on the mixing performance of the modified Tesla micromixer. The manuscript is reasonable in structure and detailed in content. However, some problems still exist. It is advised to accept them after modification.

  1. The first letter of the word “modified” in the paper title should be capitalized.

Answer> As you pointed out, the title was revised:

Mixing Performance of the Modified Tesla Micromixer with Tip Clearance

  1. Two section titles are numbered “4”, the authors should adjust them.

Answer> As you pointed out, the numbering was revised (lines 256 and 628):

  1. Results and discussion
  2. Conclusions

  1. The unit should be added for the number “65” in the demonstration of Figure 4.

Answer> As you pointed out, the caption of Figure 4 was revised (line 236):

Figure 4. Comparison of the mixing images of the first two chambers for Re = 60: (a) At z = 32.5 μm and At z = 65 μm.

  1. Not all of the sub-images in Figure 5 are numbered, and descriptions of all images are missing.

Answer> As you pointed out, the caption of Figure 5 was revised (lines 239 and 340):

(c)

Figure 5. Enhancement of the DOM by tip clearance: (a) N=3, (b) N=4, and (c) N=5.

  1. Description of sub-image (b) is missing in Figure 8 and 10, 13.

Answer> As you pointed out, the caption of figures were revised (lines 483, 538, and 609):

Figure 10. Comparison of concentration contours on the xy planes for Re = 0.1: (a) no clearance and (b) with tip clearance h = 60 mm.

Figure 12. Comparison of concentration contours on the xy planes for Re = 5: (a) no clearance and (b) with tip clearance h = 60 mm.

Figure 15. Comparison of concentration contours on the xy planes for Re = 50: (a) no clearance and (b) with tip clearance h = 60 mm.

  1. ANSYS Fluent 2021 R2 is mentioned in the line 108 and 636, but ANSYS FLUENT 2020 R2 is described in the line 168. The authors need to reconfirm the version of the ANSYS Fluent software.

Answer> As you pointed out, the version of ANSYS was revised (line 177):

ANSYS® FLUENT 2021 R2, Canonsburg, PA, USA [25]

  1. The authors should add more detailed demonstrations and simulation data to explain why tip clearance has little effects on the mixing performance when Re >50.

Answer> As you pointed out, the manuscript was revised (lines 630~646):

Figure 17 shows how the presence of tip clearance affects the vortex patterns and mixing performance in the present micromixer for the Reynolds of Re = 80. At the cross sections 1 and 2, a pair of vortices seems formed, and this is generated as the flow follows the circular passage of the micromixer; at the first and last Tesla mixing unit. It implies that the centrifugal force plays a significant role. Another interesting thing is that the vortex pattern is little affected as long as the tip clearance remains about h  60 mm. On the other hand, the vortex close to the tip clearance zone (lower vortex) is observed noticeably shrank for the tip clearance h = 70 mm; as can be seen in Figure 17 (a). This suggests that the convection flow is strong enough to localize the effects of tip clearance as long as the tip clearance is smaller than about h = 70 mm. Another pair of vortices are seen at the sections 3 and 4 which are located around the outlet of the first and last Tesla mixing units. Accordingly, the flow characteristics such as a pair of vortices plays a role as a governing mechanism in the convection dominant regime of mixing. As a result, the concentration on the plane 4 for no tip clearance and h = 60 mm is almost identical as can be seen in Figure 17 (b). Therefore, the presence of tip clearance contributes a little on the mixing enhancement for the Reynolds numbers Re  50 as long as the tip clearance remains smaller than about h = 70 mm.

  1. In the “Abstract” and “Conclusion” part, the authors show the simulation work is conducted over a wide range of the Reynolds numbers from 0.1 to 80. The authors should add simulation data and result discussion about Re around 80 in the paper content.

Answer> As you pointed out, the manuscript was revised:

In the abstract,

For the Reynolds number of Re = 80, a pair of vortices are formed around the inlet and outlet of each Tesla unit, and plays a role as a governing mechanism in the convection dominant regime of mixing. This vortex pattern is little affected as long as the tip clearance remains smaller than about h = 70 mm.

(Lines 771-775),

When the Reynolds number is larger than about 50, vortices are formed at various locations and drives the mixing in the modified Tesla micromixer. For the Reynolds number of Re = 80, a pair of vortices are formed around the inlet and outlet of each Tesla mixing unit. This vortex pattern is little affected by the presence of tip clearance as long as the tip clearance remains smaller than about h = 70 mm. It plays a role as a governing mechanism for the present micromixer in the convection dominant regime of mixing. As a result, the DOM at the outlet is little enhanced by the presence of tip clearance. The tip clearance contributes only to reduce the required pressure load for the same value of the DOM.

  1. The lateral structure mixer may be related this mixer . The author may refer: An Enhanced One-Layer Passive Microfluidic Mixer With an Optimized Lateral Structure With the Dean Effect, J. Fluids Eng. Sep 2015, 137(9): 091102.

Answer> As you suggested, the paper was referred (lines 59-60):

Various geometric modifications have been shown to generate a chaotic flow field. Some of them include a staggered herringbone [10], channel wall twisting [11], repeated surface groove and baffles [12, 13], block in the junction [14], split-and-recombine (SAR) [15, 16], Tesla structure [17], stacking of mixing units in the cross flow direction [18], optimization of lateral structure [19] and submergence of planar structures [20].

Reviewer 3 Report

The present paper reports numerical investigation of modified Tesla passive micromixer using a tip clearance parameter for a wide Reynolds number range from 0.1 to 80. The effect of Reynolds number on mixing efficiency and pressure drop is presented.

Recent research on the topic which focuses on development of novel designs for improved mixing and/or techniques to quantify mixing are packed with the most up-to-date knowledge in the field. It is difficult to find any recognizable contribution in terms of design, technique and flow physics / structures. Although a systematic analysis has been carried out, the study failed to report any advancement in the micromixing technology. Some comments to the authors:

1. Results are limited followed by a short discussion.

2. The authors must provide the information about the type of grid i.e. tetrahedral or hexahedral. Also, a quantitative validation of DOM is required in addition to qualitative comparison. For high Re number flows, the effect of numerical diffusion needs to explained both qualitatively and quantitatively.

3. Even with 5 mixing units, the performance of micromixer with tip clearance is less than 50 % for low Re flows. Therefore, one of the important requirements of obtaining lower mixing lengths is not achieved by the proposed design when compared with available designs from open literature.

4. For high Re flows, the effect of tip clearance is redundant compared to original design.

5. The paper is not well written and lacks clear communication of its contents to the readers. I strongly urge the authors to re-write the paper for effective presentation and Readability.

The manuscript is rejected mainly due to lack of novel contents and limited findings.

Author Response

Thanks for advice

Round 2

Reviewer 3 Report

All questions have been addressed.  However, the present design will require significantly long mixing lengths for complete mixing. For Re = 0.1 and N = 5, only 60% mixing is achieved, and looking at the trends in mixing with 'N', it seems N > 10 will be required for complete mixing giving very long channel lengths which can be quite impractical for micromixing applications.